# Long-Term Analysis of Aerosol Concentrations Using a Low-Cost Sensor: Monitoring African Dust Outbreaks in a Suburban Environment in the Canary Islands

**DOI:** 10.3390/s23187768

**Published:** 2023-09-08

**Authors:** Silvia Alonso-Pérez, Javier López-Solano

**Affiliations:** 1Departamento. de Ingeniería Industrial, Escuela Superior de Ingeniería y Tecnología, Universidad de La Laguna, 38206 San Cristóbal de La Laguna, Spain; 2Izaña Atmospheric Research Center, AEMET, 38001 Santa Cruz de Tenerife, Spain

**Keywords:** low-cost sensors, mineral dust, air quality, aerosols, particulate matter

## Abstract

This study presents the results of the long-term monitoring of PM10 and PM2.5 concentrations using a low-cost particle sensor installed in a suburban environment in the Canary Islands. A laser-scattering Nova Fitness SDS011 sensor was operated continuously for approximately three and a half years, which is longer than most other studies using this type of sensor. The impact of African dust outbreaks on the aerosol concentrations was assessed, showing a significant increase in both PM10 and PM2.5 concentrations during the outbreaks. Additionally, a good correlation was found with a nearby reference instrument of the air quality network of the Canary Islands’ government. The correlation between the PM10 and PM2.5 concentrations, the effect of relative humidity, and the stability of the sensor were also investigated. This study highlights the potential of this kind of sensor for long-term air quality monitoring with a view to developing extensive and dense low-cost air quality networks that are complementary to official air quality networks.

## 1. Introduction

Air pollution caused by particulate matter, which can be of anthropogenic or natural origin, has adverse effects on human health [1,2]. Various epidemiological studies conducted over the last few decades in different regions of the planet demonstrate the negative effects of exposure to high concentrations of particles, both in outdoor and indoor environments [3,4]. These effects depend, among other factors, on the particles’ size. PM10 (particles with a diameter less than 10 μm) have a negative impact on a significant portion of the respiratory system, causing irritation of the airways and affecting individuals with asthma and rhinitis, among others. As for PM2.5 particles (with a diameter less than 2.5 μm), due to their small size, they can penetrate deep into the lung alveoli and even enter the bloodstream, leading to respiratory diseases, cardiovascular diseases, and premature death. It is estimated that the worldwide premature mortality rate due to exposure to high concentrations of PM2.5 is 253 deaths per million inhabitants [5]. During the year 2015, high concentrations of PM2.5 caused at least 4.3 to 4.4 million premature deaths worldwide, with this number potentially reaching 7.7 million, primarily due to ischemic heart diseases [6]. Besides the effect on human health, particulate matter may have a noticeable impact on the economy of the affected areas, for example by causing visibility problems which might affect the transportation of goods and citizens or affecting the influx of tourists to areas impacted by high PM levels [7].

One important source of natural particulate matter in the air is the mineral dust injected into the atmosphere by arid and semiarid soils. The particulate matter in the source regions is injected into the atmosphere when favorable meteorological and soil conditions occur, reaching heights of up to 8 km [8,9]. Then, the mineral dust is transported by winds and can travel thousands of kilometers from its source.

The proximity of the Canary Islands to significant dust source areas in the African continent results in frequent and intense African dust outbreaks coinciding with east, south, or southeast winds. The mineral dust arriving at this archipelago primarily originates from the Western Sahara, Algeria, Mauritania, and Mali regions [10]. The meteorological pattern that most frequently gives rise to these situations throughout the year consists of a high-pressure system over Europe affecting northern Africa. In some cases, this system becomes elongated and intense enough to form a high-pressure ridge, with the Canary Islands located on the southwestern flank of this high-pressure system. Other meteorological scenarios responsible for the transport of African dust toward the Canary Islands include the combination of a low-pressure center, which can be located either northeast or southeast of the islands, with high-pressure systems in the Mediterranean and/or northern Africa [11,12,13]. In summer, the export of African dust to the Atlantic, and its pulsating behavior, is modulated by the interaction between the dipole formed by a high-pressure system over the subtropical Sahara and a low-pressure system over tropical North Africa, and mid-latitude Rossby waves [14,15,16]. 

Various characteristics of dust particles in the air can be studied, including the number of particles, concentration in different size ranges, chemical composition, size, shape, absorption coefficient, dust layer height, and optical thickness of aerosols due to dust. Some of these measurements are performed in-situ from observatories, while other measurements are carried out using instruments aboard artificial satellites. The in-situ measurements of dust concentration hold significant importance as they form the basis for epidemiological studies. These measurements serve as the benchmark for formulating regulations and recommendations aimed at mitigating the adverse effects of these particles on human health. Different types of instruments are employed to measure airborne particle concentrations. Some of these instruments are pieces of reference equipment used for the continuous monitoring of dust concentrations to ensure compliance with current regulations (for instance, high volume samplers, beta radiation attenuation monitors, and tapered element oscillating microbalance instruments). Furthermore, there are alternative measurement instruments that, while not classified as reference or equivalent devices, offer distinct advantages and prove to be valuable for air quality monitoring in specific applications.

In the past decade, there has been a growing interest in miniaturizing and reducing the cost of dust concentration sensors, making them accessible even to citizens participating in citizen science projects. Most of these sensors rely on the physics of light scattering, primarily using visible and near-infrared light. Light scattering is affected by particle size, thus these devices are capable of measuring different particle fractions, including PM10, PM2.5, and PM1 (the former two are relevant for African dust studies, while the very fine PM1 is mainly associated with anthropogenic emissions). These sensors have prices in a large range, from tens to hundreds of thousands of dollars. Numerous manufacturers offer various models in the market, some of which have been found to be more reliable than others based on multiple data analyses conducted by the scientific community [17,18]. Users have the option to either build a low-cost monitor using these sensors, plus any other necessary components for control and data acquisition, or purchase a fully operational device at a higher price. Extra sensors can also be included to monitor other variables, such as temperature, relative humidity, UV radiation, and even ambient noise [19]. The measurements made by low-cost dust concentration sensors, at least at present, are not as reliable as those made by the reference instruments used by public administrations to assess compliance with current air quality regulations. Low-cost sensors have a shorter lifespan compared with more expensive systems, and their measured values depend heavily on environmental conditions, especially humidity [17]. Therefore, it is crucial to compare and calibrate these instruments against reference or equivalent equipment, preferably under the same operating conditions [17].

However, despite the disadvantages of low-cost sensors compared with more sophisticated instruments, the possibilities of miniaturization, reduced cost, simple maintenance, and low energy consumption make them very useful for certain applications that do not require the precision necessary for full scientific data usage. These low-cost sensors are suitable for detecting spikes in air pollution and monitoring the exceedance of concentration thresholds, thus they can be used to issue specific alerts. In addition, their characteristics make them ideal for deploying extensive and dense air quality networks which provide both higher spatial and temporal resolution than official air quality networks. There are currently numerous citizen science projects that utilize this type of sensor, providing real-time data that is accessible to any citizen with an internet connection. An example of this is the global sensor network Sensor.Community [20] which has sensors deployed in five continents, mainly in Europe. Our low-cost sensor is part of this network.

Some studies have been carried out to assess the impact of mineral dust outbreaks or local dust storms on PM10 and PM2.5 concentrations using low-cost sensors [21,22]. However, to the extent of the authors’ knowledge, none of them have accomplished taking continuous measurements for more than 2 years. In the present study, we analyze continuous measurements made during the period of 10 November 2019–28 May 2023 with a Nova Fitness SDS011 sensor. The life service of this sensor is estimated by the manufacturer to be less than 1 year [17]. Further, the sensor was located in an urban area frequently affected by African dust intrusions, and with high relative humidity. Minimal maintenance has been performed, pushing the sensor to the limit to analyze its performance over time and assess its drift with respect to the detection of African dust intrusions. It must be noted that these conditions could be the same found in a citizen science project or in any semi-autonomous network operating in a remote site with low accessibility.

## 2. Materials and Methods

### 2.1. Low-Cost Dust Monitoring System

The particulate matter low-cost sensor used in this study was an SDS011, manufactured in Jinan, China by Nova Fitness Co., Ltd. We chose this sensor because it is one of the most popular ones in low-cost dust sensor studies and in citizen science networks. This sensor is suitable for taking PM10 and PM2.5 measurements both indoors and outdoors. Its measurement technique relies on the physics of light scattering, using a laser beam which radiates light into the air containing dust particles. Then, a photodiode detects the amount of this light which is dispersed by the particles suspended in the air. Finally, a microcontroller unit calculates the PM10 and PM2.5 concentrations using the information given by the photodiode. The resolution of this sensor is 0.3 μm, its accuracy is ±15% of the measurement (minimum of ±10 μg/m^3^), and its measurement range is from 0 to 1000 μg/m^3^. According to the manufacturer, it can operate under temperatures ranging from −10 °C to 50 °C, and under relative humidity ranging from 0 to 70% [17]. At the time of its acquisition, the price of this sensor was approximately 16 USD.

To perform temperature and relative humidity measurements, we used a DHT22 (AM2302) sensor manufactured by Aosong Electronics in Guangzhou, China. The price of this sensor was around 0.5 USD. According to the manufacturer, this sensor can measure temperatures from −40 °C to 80 °C with an accuracy of ±0.5 °C, and relative humidity from 0% to 100% with an accuracy of ±2% [23]. The DTH22 sensor did not operate for the entire duration of the experiment and had to be replaced two times. We also measured barometric pressure using a BMP180 sensor, although the pressure data are not used in this work.

The signal from the PM, temperature/humidity, and barometric pressure sensors is received by a NODE MCU v3 microcontroller manufactured by Espressif Systems in Shanghai, China. This microcontroller, with a CH341 chipset, is a development board based on the ESP8266 Wi-Fi microchip. Its price was approximately 1.6 USD. We programmed the microcontroller with firmware provided by Sensor.Community [24]. We configured the system to take one measurement every 5 min. 

We housed the sensors and the microcontroller in a plastic enclosure, in which we drilled one hole to insert a PVC tube 9 cm in length and 6 mm in diameter connected to the SDS011 air inlet. Note: Sensor.Community recommends a 20 cm inlet [25], but we have used an even shorter one to minimize particle losses via impact. Six holes of 5 cm diameter were drilled into the opposite side of the enclosure to allow airflow. The low-cost dust monitoring system assembly, as well as the diagram of connections of the sensors to the Node MCU, are shown in Figure 1.

Our low-cost dust monitoring system is installed in a location with no direct power connection, so it was powered by a 25,000 mAh power bank with a Li-polymer battery, the Soluser F-TS-819, which includes a solar panel. This battery is manufactured by Soluser-UK in China. The battery is positioned above the plastic enclosure, conveniently accessible to facilitate battery replacement, while a USB cable connects the battery to the USB port of the Node MCU. We found it necessary to replace the battery approximately every 6 days, with this duration gradually decreasing over time to around 4 days in May 2023. Although we only used one battery at the beginning of the experiment, we later used two, so that while one battery was charging, the other was in use in the sensor. To determine when to replace the battery and minimize data gaps, we implemented a notification system that sent a message to our smartphones when a computer failed to connect via Wi-Fi to the dust monitor. This computer, a low-cost Raspberry Pi 3 manufactured by the Raspberry Pi Foundation, also runs our own Python scripts to gather the data from the monitor every 5 min and automatically produce plots to check the daily evolution of the PM10 and PM2.5 values.

As mentioned, minimal maintenance was performed in our monitoring, as one of our objectives was to study the lifetime of the dust sensor, its performance over time, whether there is any drift with respect to detection of African dust intrusions, and the degradation of the components. Only the DTH22 sensor and the USB power cable connecting the SDS011 to the external power bank had to be replaced during the experiment.

### 2.2. Study Sites: Low-Cost Sensor and Reference Monitor

Our low-cost sensor operated in San Cristóbal de La Laguna, the second largest city on the island of Tenerife. Tenerife is the largest of the Canary Islands and is located in the Atlantic Ocean off the northwest coast of Africa. This archipelago exhibits a subtropical climate characterized by mild temperatures throughout the year and relatively stable weather patterns. This climate is influenced by the trade winds. When the trade winds are absent, and winds are predominantly coming from the east or southwest direction, the occurrence of Saharan dust outbreaks is common [9,10].

Our low-cost dust sensor was located outside a third-floor window in a four-story residential building facing a suburban street, at latitude 28°29′24.0″ N, longitude 16°18′36.0″ W, and height 558 m above sea level (a.s.l.) (Figure 2). The airborne dust levels at this site are mainly influenced by road traffic and occasional African dust intrusions since there is no nearby industrial activity or other sources of dust. Road pollution can also be considered fairly low, the closest road has low traffic, and the measurement site is close to green areas.

For the comparison of PM10 and PM2.5 levels with a reference dust monitor, we chose the nearest reference air quality station, Vuelta de Los Pájaros (VP) (latitude 28°27′43.19″ N, longitude 16°16′37.13″ W, height 168 m a.s.l.) (Figure 2). This station, based on an Envea MP101M beta attenuation monitor, is owned by the company CEPSA and is part of the air quality network of the Canary Islands Government. The distance from our low-cost sensor to the VP monitoring station is 4.9 km. The PM levels at VP are strongly influenced by road traffic and emissions from the brewery located at the same site. 

Note that the PM10 and PM2.5 values of the low-cost and reference sensors are not expected to be the same, because the two instruments are not collocated, and their calibrations are presumably not related. Some degree of correlation between the data of both sensors can, however, be expected, because African dust outbreaks affect large areas. In this paper, we use the reference instrument to detect changes in the low-cost sensor, either over time or caused by the relative humidity. This is accomplished by looking at changes in the differences or in the correlations between both datasets. The reference instrument is also useful for helping in the identification of the days affected by African dust.

### 2.3. Data Analysis

We have performed 5-min measurements of PM10, PM2.5, relative humidity, and temperature with our low-cost sensors almost continuously during the period from 10 November 2019 to 28 May 2023—1020 days in total. Then, daily values were calculated using the median to avoid the effect of outliers.

For the identification of high PM10 levels, we used a 50 μg/m^3^ threshold, which corresponds to the daily PM10 value that cannot be exceeded more than 35 times in a year according to the European Air Quality Directive 2008/EC/50 [26]. In the Canary Islands, PM10 concentrations are usually below this value, and the exceedances of this daily threshold occur almost exclusively during African dust episodes [14] and references therein. Moreover, this PM10 threshold coincides with the daily PM10 value associated with heart failure in-hospital mortality during African dust outbreaks in the Canary Islands [27,28]. Because the factory calibration of our sensor might present some offset, a sensibility analysis was done to determine if there is another daily PM10 threshold that better identifies African dust days.

The comparison of daily time series of PM10 and PM2.5 measured on both sites was performed through correlation analysis, mean absolute error (MAE) (Equation (1)), and root mean square error (RMSE) (Equation (2)):(1)$$MAE=\frac{\sum_{i=1}^{n}\left({PM}_{ref}-{PM}_{lowcost}\right)}{n}$$
(2)$$RMSE=\sqrt{\frac{\sum_{i=1}^{n}{\left({PM}_{ref}-{PM}_{lowcost}\right)}^{2}}{n}}$$

In both Equations (1) and (2), PM_ref_ and PM_lowcost_ are the particulate matter concentrations measured in the reference monitor and in the low-cost sensor, respectively, and n is the number of observations. To examine the presence of a linear relationship between two variables, the coefficient of determination based on the Pearson correlation coefficient was calculated. Furthermore, to analyze the potential existence of a monotonic relationship between two variables that may exhibit some outliers, the Spearman correlation coefficient was also calculated. In both cases, *p*-values were calculated to determine the statistical significance of the results.

### 2.4. Assessment of African Dust Intrusions

For the identification of African dust intrusion days, we consulted the annual reports on episodes of high particle concentrations due to natural events published by the Ministry for Ecological Transition and Demographic Challenge of Spain (METDC) [29]. It should be mentioned that, at the time of conducting this study, these annual reports were only available up to 2021. Although a list of validated African dust days is not yet available for years 2022 and 2023, provisional data are available online.

These reports are carried out following the methodology described in [30], which is based on the analysis of meteorological synoptic scenarios, air-mass back-trajectories, and particulate concentration measurements, reanalysis of dust models, and satellite images. Using these data, the reports are able to identify the days in which the entire Canary Islands are affected by African dust.

## 3. Results and Discussion

### 3.1. Identification of African Dust Outbreaks

#### 3.1.1. PM10 Concentrations Due to African Dust Outbreak

Figure 3 shows yearly PM10 plots of both our low-cost sensor and the VP dust monitor. A good correlation is already evident from these plots. Note that there are some periods where data from the low-cost sensor are missing. This is especially the case at the beginning of the experiment, when we faced problems keeping the sensor continuously powered, causing data gaps of up to approx. 9 days, although usually of just 1–2 days. As described in Section 2.1, this issue was solved by using two batteries in rotation and implementing a warning system in our Raspberry Pi, which helped us to shorten the data gaps to 1–2 days in most cases. Other periods without data correspond to the malfunctioning of the humidity-temperature sensor. This was the cause for the ~1.5-month period without data from early March to mid–April 2020. We later decided to keep the SDS011 sensor running even when we were waiting for a replacement of the DHT22 sensor to arrive.

As it can be seen in Figure 3, our low-cost sensor is measuring, almost systematically, lower values of daily PM10 concentrations than the VP dust monitor. Only on 5.4% of the days did the low-cost sensor measure values higher than those measured by VP.

According to the information provided by the METDC about episodes of high particle concentrations due to natural events, 527 days of African dust outbreaks occurred in the Canary Islands during our entire study period (Table 1). Of course, as an African dust intrusion may not necessarily impact all islands within the Canary Islands, or it may affect different altitudes within the same island, not all the episodes shown in Table 1 necessarily affected our study area. In fact, if we analyze our complete data time series (10 November 2019–28 May 2023), n = 995 days with data in both sites, and VP registered 118 days with PM10 above 50 μg/m^3^, with 116 of them (98.3%) being among those shown in Table 1. On the other hand, our low-cost sensor registered 53 days with PM10 above 50 μg/m^3^, with 51 of them being in Table 1.

However, our SDS011 sensor is working with the calibration provided by the manufacturer, which might not be correct, thus the 50 μg/m^3^ threshold for the determination of African dust outbreaks might not be optimal. We have performed a sensibility analysis to determine the optimal daily PM10 threshold for our low-cost sensor to identify the maximum number of African dust intrusions comparing with the METDC information. We found that the optimal threshold was 37 μg/m^3^, since 100% of the days with PM10 > 37 μg/m^3^ (73 days) correspond with African dust intrusion days.

There is still a noticeable difference between the number of days identified by the VP and low-cost sensors: 116 vs. 73. Part of this difference can be attributed to the different location of both sites—note that, besides both sites being 5 km away, there is a difference of 400 m in height, and the orography of the surrounding areas is completely different (see Figure 2). The placement of the SDS011 sensor is representative of a citizen science experiment, but not optimal for measurements, because one side is completely blocked by a wall. 

Of course, it might also be that the SDS011 sensor is producing completely different data than the VP sensor. Low-cost sensor daily PM10 concentrations for the African dust days with PM10 > 37 μg/m^3^ in our low-cost sensor range between 37.1 and 853.8 μg/m^3^, while for clean days they range between 0 and 268 μg/m^3^. Two spikes of PM10, with 108.4 μg/m^3^ on 7 August 2020 and 268.4 μg/m^3^ on 5 October 2020, are observed for days without African dust influence. The reference dust monitor in VP did not detect high PM10 concentrations on those days, so we can guess that these peaks could be due to some malfunctioning of our low-cost sensor (for example, a data read, or transmission, error), or due to a local source of anthropogenic dust pollution. Examining the daily PM10 and PM2.5 data (see Section 3.1.2), we concluded that these peaks were most likely caused by a malfunctioning of the sensor. If we remove those two days, PM10 values during clean days range between 0 and 44.6 μg/m^3^. We will examine the SDS011 and VP measurements in more detail in Section 3.2. 

#### 3.1.2. PM2.5 during African Dust Intrusions

For the entire study period, the correlation between daily PM2.5 and PM10 measured with our low-cost sensor was very strong, with a Spearman coefficient of r = 0.91 (*p*-value < 0.001). The coefficient of determination from a linear regression between PM2.5 and PM10 was also obtained, resulting in R^2^ = 0.58 (*p*-value < 0.0001). A high correlation between the two values is expected, since both are calculated internally by the SDS011 sensor using the same data. Figure 4 shows that PM2.5 values measured with our low-cost sensors increase during the African dust outbreaks. Daily PM2.5 concentrations during the dust outbreaks with PM10 > 37 μg/m^3^ in our low-cost sensor range from 4.7 to 767.0 μg/m^3^.

As in the case of PM10 mentioned above, two spikes of PM2.5 were observed on 7 August 2020 and 5 October 2020, with 101.1 μg/m^3^ and 246.8 μg/m^3^, respectively. These values are very similar to those of PM10 for the same days, with PM2.5/PM10 ratios of 0.93 and 0.92 for 7 August 2020 and 5 October 2020, respectively, which indicates a fine granulometry that is not usual for African dust intrusions reaching the Canary Islands (PM2.5/PM10 ratio of approx. 0.3) [31]. This unlikely situation reinforces the possibility of some malfunctioning of the low-cost sensor on these days. If we remove these two dates from the statistics, PM2.5 concentrations during clean days range between 0 and 10.3 μg/m^3^.

### 3.2. Performance of the Low-Cost Sensor

#### 3.2.1. Daily PM10 Comparison with the Reference Dust Monitor

We performed a comparison of daily PM10 values from the low-cost sensor and the reference monitor in VP for our entire study period, totalling 991 days with data from both instruments. Four days with values of PM10 above 100 μg/m^3^ measured with the low-cost sensor and under 20 μg/m^3^ at VP were removed from the time series. Two of those days coincide with the 7 August 2020 and 5 October 2020 previously identified with a malfunctioning of the low-cost sensor. The remaining two days were 28 July 2020 and 23 February 2020, days with very intense African dust outbreaks in the Canary Islands but no data from the VP monitor (the dust might have saturated this monitor, or the device might have experienced some problem). Excluding these days, a linear regression of the low-cost vs. VP PM10 values resulted in R^2^ = 0.78 (*p*-value < 0.0001) for the coefficient of determination (Figure 5). To avoid the effect of outliers, the Spearman coefficient of correlation was also calculated, resulting in a value of r = 0.76 (*p*-value < 0.001). As previously seen in Figure 3, the low-cost sensor generally underestimates daily PM10 values compared with those of the reference sensor.

For this daily PM10 comparison, an MAE = 15.4 μg/m^3^ and an RMSE = 33.7 μg/m^3^ were obtained. Note that the MAE value is the difference between the average PM10 of the VP sensor (which we assume is correctly calibrated and thus produces correct values) and our SDS011 sensor (which is running with the calibration provided by the manufacturer). Then, the MAE value can be compared with the result of our sensitivity analysis in Section 3.1.1, in which we concluded that 37 μg/m^3^ is the threshold for the detection of African dust breaks in our sensor. Compared with the standard threshold of 50 μg/m^3^, we find a difference of 13 μg/m^3^, which is indeed fairly close to the MAE value of 15.4 μg/m^3^. The MAE value is also comparable to those found by Tagle et al. [32] for SDS011 low-cost sensors compared with a reference instrument in Santiago de Chile, especially with the sensor exposed to the highest average relative humidity during their campaign. Finally, note that the high RMSE value suggests that our low-cost sensor requires proper calibration.

Examining the data in more detail, a shift in the absolute error seems to be observed in Figure 6 starting in February 2022. To verify the presence of this drop, MAE for the periods 10 November 2019–31 January 2022 and 1 February 2022–28 May 2023 were calculated, obtaining values of MAE = 15.9 μg/m^3^ and MAE = 14.7 μg/m^3^, respectively. Hence, our findings indicate that, since February 2020, the low-cost sensor has been measuring daily PM10 values 1.2 μg/m^3^ higher than before. It is worth noting that this change happened after an intense African dust outbreak, so that the high PM levels might have affected the sensor.

To study the effect of relative humidity on the daily PM10 values measured with our low-cost sensor, we performed the correlation analysis with n = 556 days, when data from both the SDS011 dust sensor and the DHT22 relative humidity sensor are available. Repeating the previous analysis for this smaller dataset, we still found a good linear relationship between the daily PM10 values (Figure 7). The relative humidity for this dataset ranged between 1% and 95%, with a mean value of 60.7%. Note that some relative humidity values exceeded the 70% maximum recommended by the manufacturer, as the study site is located in a city characterized by high humidity throughout the year. Daily median values above 70% relative humidity happened for 219 days, 39.4% of the total. We found that relative humidity had no clear effect on the data for our site (see Figure 7), in contrast with other authors who found that relative humidity above 80% [33] or 95% [34] negatively affected the SDS011 sensor response. In our dataset, values above the 80 and 95% relative humidity thresholds were found in 17.3% and 0.2% of the analyzed days, respectively. Note that the Spearman correlation coefficient between PM10 and relative humidity is just 0.20 (Pearson coefficient of 0.093).

#### 3.2.2. Daily PM2.5 Comparison with the Reference Dust Monitor

For the comparison of the daily PM2.5 values measured with the low-cost sensor and the VP reference monitor, we removed the data for 23 February, 28 July, 7 August, and 5 October 2020, as we again found (see Section 3.2.1) outliers in these days, likely due to the malfunctioning of either the low-cost sensor or the reference dust monitor. After removing these four points, the dataset still contains 976 days, and we found a strong linear relation, with a correlation coefficient of R^2^ = 0.76 (*p*-value < 0.01) and a Spearman coefficient of r = 0.78 (*p*-value < 0.001). Figure 8 shows that, as in the case of the daily PM10, the low-cost sensor tends to underestimate daily PM2.5 values in comparison with the values measured by the reference monitor.

For the daily PM2.5 comparison, an MAE = 6.05 μg/m^3^ and an RMSE = 12.4 μg/m^3^ were obtained. These errors are lower than in the case of PM10. Holder et al. [35] have demonstrated that low-cost sensors can effectively address spatial gaps of reference networks in hourly PM2.5 temporal series, even near wildfires, when they achieve an MAE < 10 μg/m^3^. The RMSE value is lower than that found by Zheng et al. [36] using a Plantower PMS7003 low-cost sensor at a daily scale, although it suggests that calibration is needed for our PM2.5 measurements.

To verify if, as in the case of PM10, there is a jump in the MAE from February 2022 onward, this error was calculated for the periods 10 November 2019–31 January 2022 and 1 February 2022–28 May 2023, obtaining values of MAE = 6.5 μg/m^3^ and MAE = 5.3 μg/m^3^, respectively. Hence, our findings indicate that, since February 2020, the low-cost sensor has been measuring daily PM2.5 values around 1.2 μg/m^3^ higher than before. This drop in PM2.5 measurements is the same as in PM10.

Finally, we studied the effect of relative humidity on the daily PM2.5 values measured with our low-cost sensor. To accomplish this, we performed the correlation analysis with n = 553 days when PM2.5 data are available for both dust sensors and the DHT22 relative humidity sensor. As in the case of PM10, Figure 9 shows a good linear relationship between the daily PM2.5 values with no clear effect of the relative humidity. The relative humidity for this dataset, as in the case of Figure 7, ranges between 1% and 95%, with a mean value of 60.6%. For the PM2.5 dataset, the number of days above the 70%, 80%, and 95% relative humidity is almost the same as for the PM10. Moreover, as in the case of the correlation of PM10 on both sites, we found that relative humidity had no clear effect on the PM2.5 data. The Spearman coefficient between PM2.5 and relative humidity is again very low, at 0.15 (Pearson coefficient of 0.094).

## 4. Conclusions

We have analyzed almost 3 and a half years of data measured with a low-cost SDS011 sensor operating in a suburban area occasionally affected by African dust outbreaks. We have found that there is a high correlation between the PM10 and PM2.5 data of our low-cost sensor and those of a reference instrument operating 5 km away. Using the factory calibration, we find that our low-cost sensor provides PM10 values approx. 15 μg/m^3^ lower than the reference instrument. Still, taking the effect of the calibration into consideration, we find that the low-cost sensor can successfully detect African dust outbreaks. Differences in the identification of African dust days with the reference instrument can at least partially be attributed to the differences in the measurement sites, but a more detailed analysis of specific events is required.

Over the period considered, the low-cost sensor has been fairly stable, with just one event which caused a small shift in the comparison with the reference instrument for both PM10 and PM2.5. Besides a 1.5-month period at the beginning of the experiment, our low-cost sensor has operated almost continuously over the 3.5-year period presented in this paper, so the instrument can be also considered fairly robust. Contrary to previous studies, we did not observe a noticeable effect of relative humidity on the low-cost sensor. It is worth stressing that we used the low-cost sensor as provided by the factory, without further calibration and with minimal maintenance. The firmware used in our SDS011, supplied by the Sensor.Community network, already implements automatic submission of the data, and we have further automatized basic data analysis through the generation of plots with daily data, and implemented a warning system for power loss events.

All the above means that these kinds of low-cost instruments could be used in large, semi-autonomous networks, perhaps operated by citizen scientists with minimal training. These networks could supplement official networks, providing additional data with a high spatial and temporal resolution. This is especially important in areas with a complex orography, such as the Canary Islands. Given the impact of particulate matter on health, these low-cost networks could be immediately useful to the local citizens.

With a view to further understand the differences between the low-cost and reference sensors, besides the aforementioned analysis of specific events, we also intend to perform experiments with both types of instruments operating at the same site. Note also that, although from our present results we can already determine how to improve the calibration for our SDS011 low-cost sensor, this experiment with both instruments collocated will also provide further data to improve this calibration. Further future work also includes performing experiments running multiple SDS011 low-cost sensors at the same time, with the objective of studying the differences among them and the possibility of obtaining better results from the combination of their measurements. This type of experiment would also allow us to check the effect of sensor placement on the measurements. In order to improve the autonomy of the sensor and reduce even more the need for human intervention, we would also like to investigate methods to reduce energy consumption, such as switching the Node MCU to a low power state between measurements or using LoRa or Bluetooth Low Energy technologies for data transmission.

## Figures and Tables

**Figure 1 sensors-23-07768-f001:**
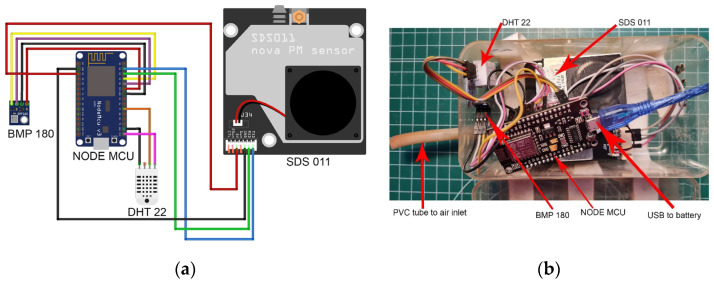
(**a**) Diagram of connections, and (**b**) our low-cost dust monitor after 3 years and 5 months of continuous operation.

**Figure 2 sensors-23-07768-f002:**
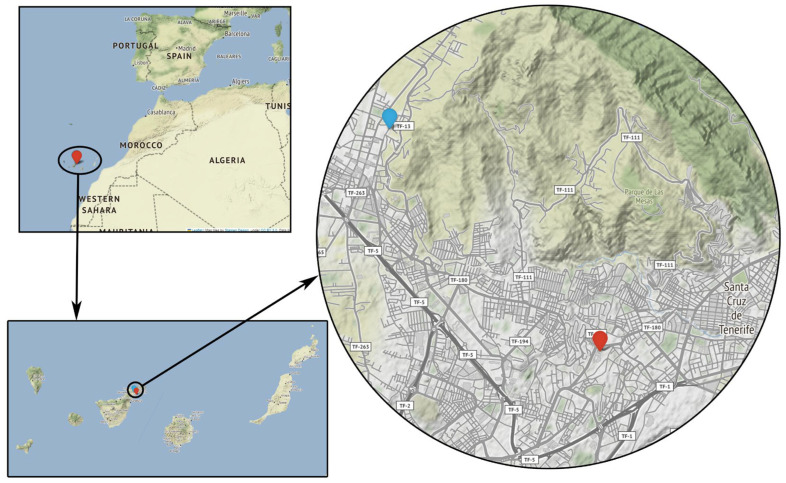
Location of our low-cost dust sensor (blue mark) and Vuelta de Los Pájaros (red mark) reference station.

**Figure 3 sensors-23-07768-f003:**
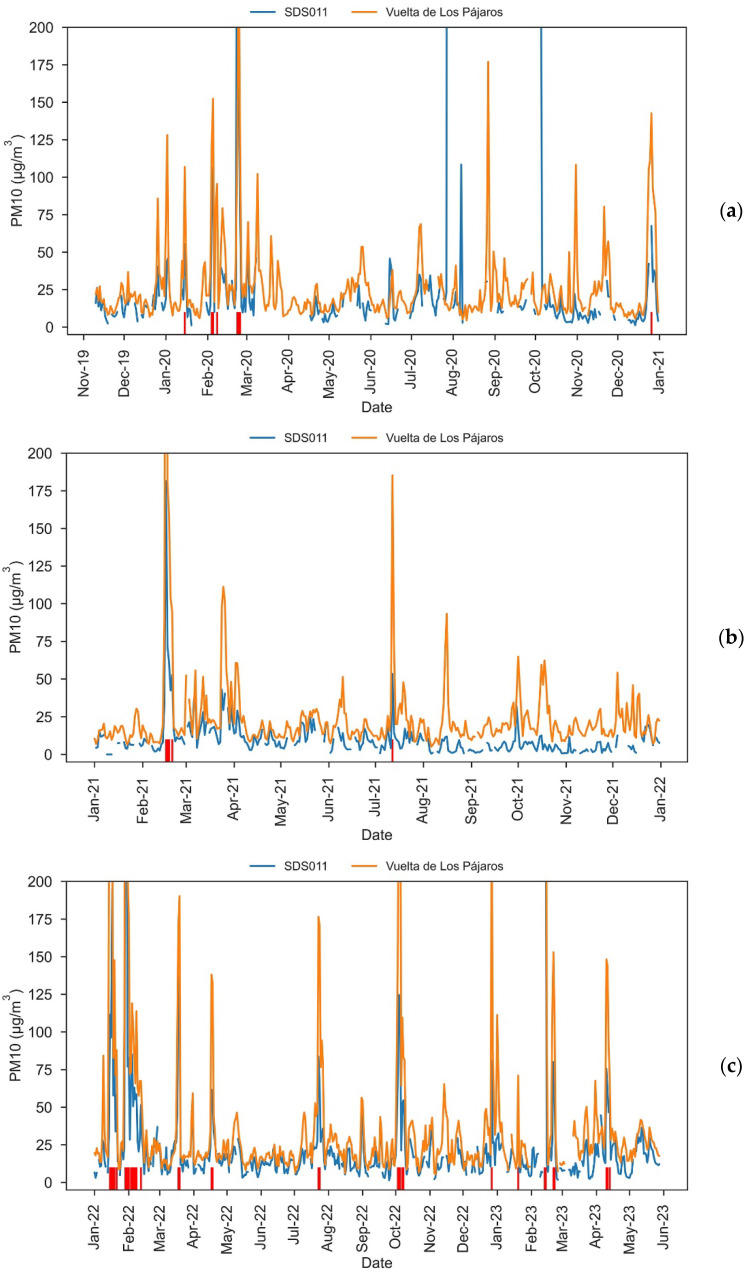
Daily PM10 measured with our SDS011 low-cost sensor (blue) and with the Vuelta de Los Pájaros reference monitor (orange) during the periods (**a**) 10 November 2019–31 December 2020, (**b**) 1 January 2021–31 December 2021, and (**c**) 1 January 2022–28 May 2023. Red vertical lines indicate days in which both sites were affected by African dust.

**Figure 4 sensors-23-07768-f004:**
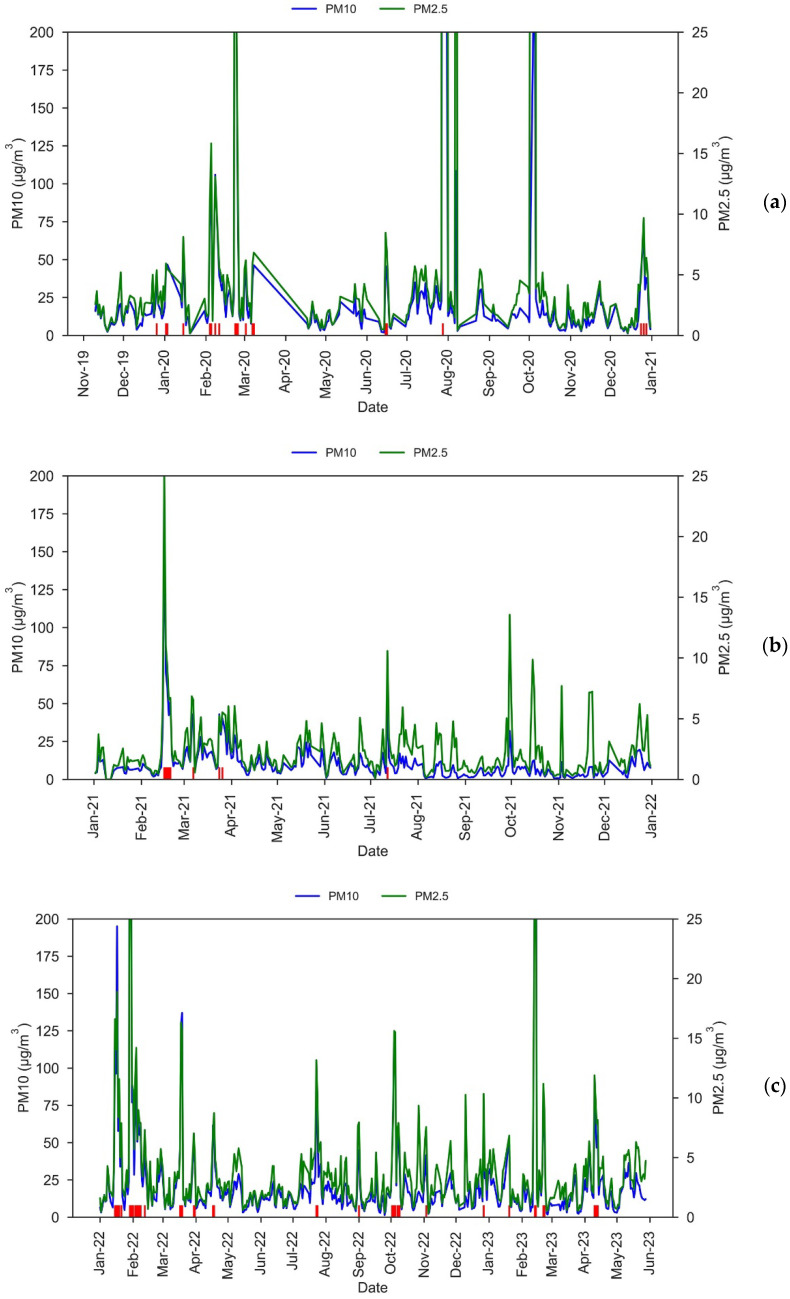
PM10 (blue) and PM2.5 (green) daily values measured with the low-cost dust sensor during the periods (**a**) 10 November 2019–31 December 2020, (**b**) 1 January 2021–31 December 2021, and (**c**) 1 January 2022–28 May 2023. Red vertical lines indicate days with PM10 > 37 μg/m^3^ which are African dust days according to METDC.

**Figure 5 sensors-23-07768-f005:**
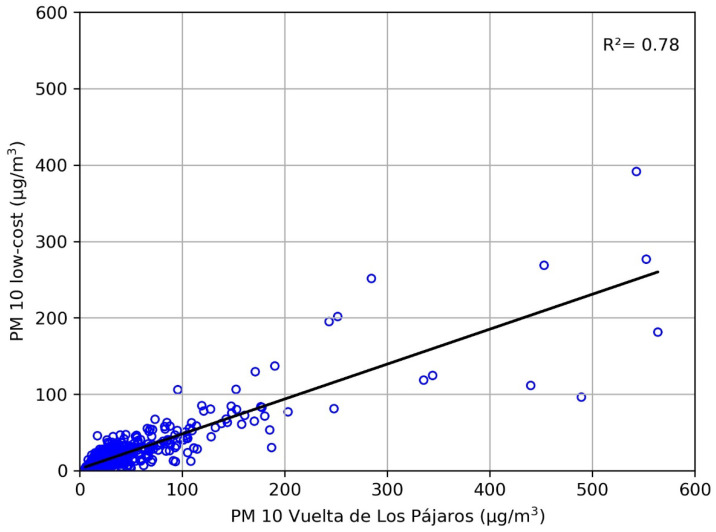
Correlation of daily PM10 concentrations at the low-cost sensor and the Vuelta de Los Pájaros reference monitor.

**Figure 6 sensors-23-07768-f006:**
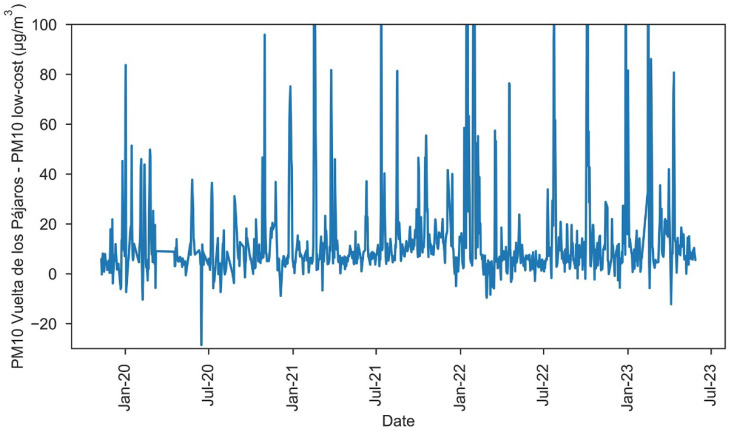
Absolute error of Daily PM10 measured by the low-cost sensor and by the Vuelta de Los Pájaros reference monitor.

**Figure 7 sensors-23-07768-f007:**
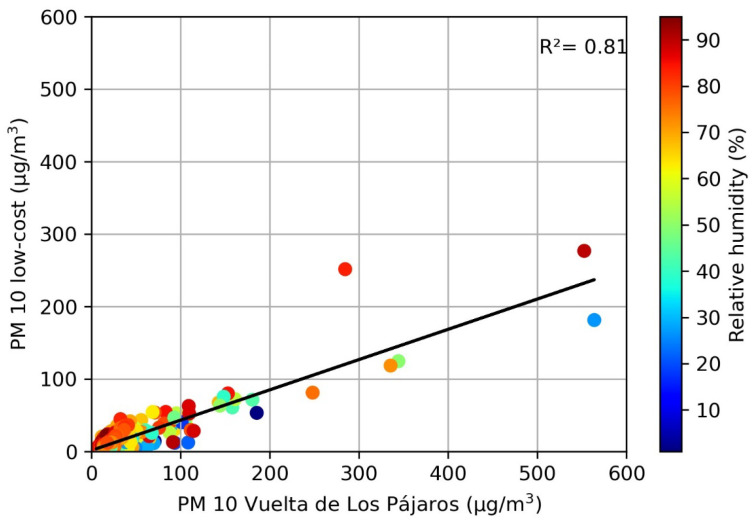
Correlation of daily PM10 concentrations at the low-cost sensor and the Vuelta de Los Pájaros reference monitor. The relative humidity measured with the low-cost DHT22 sensor is shown in different colors.

**Figure 8 sensors-23-07768-f008:**
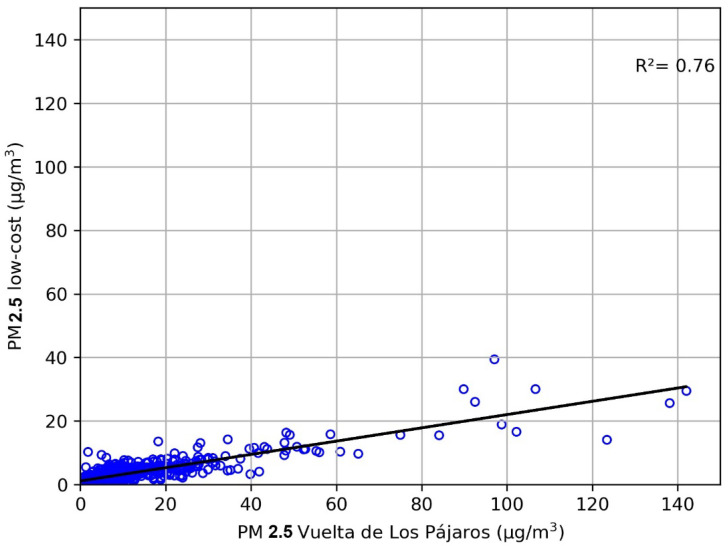
Correlation of daily PM2.5 concentrations at the low-cost sensor and the Vuelta de Los Pájaros reference monitor.

**Figure 9 sensors-23-07768-f009:**
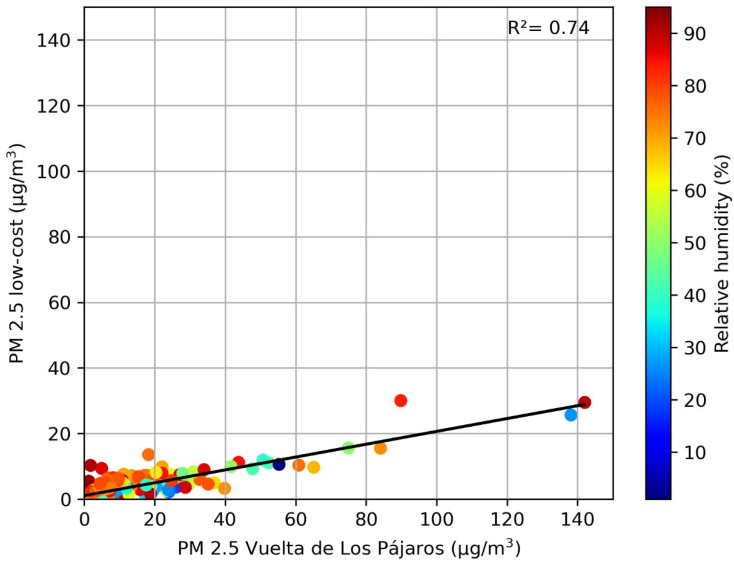
Correlation of daily PM2.5 concentrations at the low-cost sensor and the Vuelta de Los Pájaros reference monitor. The relative humidity measured with the low-cost DHT22 sensor is shown in different colors.

**Table 1 sensors-23-07768-t001:** Validated and provisional dates of African dust intrusion periods in the Canary Islands, from the start of our study period to December 2021 (validated), and to May 2023 (provisional) (METDC).

Dates of African Dust Episodes	Month	Dates of African Dust Episodes	Month
11–13; 26–30	November 2019	18; 27–30	September 2021
1–3; 14–22	December 2019	1; 8–10; 23–31	October 2021
6–7; 21–23; 29	January 2020	1–4; 12–18; 27–31	November 2021
4–7; 10–21; 29–31	February 2020	1–20; 22–29	December 2021
1–4; 8–22; 26–31	March 2020	1–2; 5–13; 19–20; 24–27	January 2022
1–14; 16–18; 22–28	April 2020	No dust episodes	February 2022
1; 17–20; 30–31	May 2020	3; 19–31	March 2022
3; 9–10; 16–19	June 2020	15–17	April 2022
1–2; 7–12; 22	July 2020	31–1	May 2022
18–19	August 2020	1–3; 22–31	June 2022
8–11; 17–27	September 2020	1–2; 6–10; 23–27	July 2022
1–8; 20–22; 30–31	October 2020	7–8; 11–14; 19–20; 29–31	August 2022
1–2; 11; 17–18	November 2020	1–4; 14–26	September 2022
2–10; 13–31	December 2020	22–30	October 2022
1–3; 7–8; 13–17; 21; 26–27	January 2021	13–14; 18–20; 27–30	November 2022
19–31	February 2021	15–20; 25; 28	December 2022
1–8; 12; 14; 20	March 2021	1–4; 6–7; 9–12; 23–31	January 2023
1–6; 10–14; 18–24	April 2021	8–1	February 2023
12–13; 21–31	May 2021	16–19; 20–25	March 2023
1; 7–14; 27	June 2021	12–7	April 2023
No dust episodes	July 2021	21–11	May 2023
	August 2021	14–19	

## Data Availability

The data presented in this study are available on request from the corresponding author.

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
