# Peer review of "Long-Term Analysis of Aerosol Concentrations Using a Low-Cost Sensor: Monitoring African Dust Outbreaks in a Suburban Environment in the Canary Islands"

_sensors, 2023, doi:10.3390/s23187768_

Round 1

Reviewer 1 Report

The authors present another monitored site in the world where, in addition, the impact of African dust on the air in the Canary Islands can be monitored. These are all useful pieces of information leading to an understanding of the occurrence of PM in individual locations. I wish the authors great success in meeting the objectives stated from line 446 onwards in the form of further publications.

My comments are following:

- In Fig.1, let the 3 lines be perpendicular, not inclined.

- What is the measurement accuracy of SDS011? Please also provide this information in ch.2.1, please.

- Does the design of the sensor require a PVC tube? Can the sensor be mounted directly in the air? I think the PVC tube may potentially distort the results in Fig.3.

- How many power banks were needed for the measurement? 2 pcs or more?

- What tool was used to automatically produce plots to check the daily evolution of the PM?

- Describe in more detail the work with the Raspberry Pi 3 when measuring the PM.

- I recommend to put web links in References.

- What were the causes of the measurement failures with SDS011 on Fig.3?

- How long were the measurement interruptions with SDS011 (hours, days)?

- I recommend putting Fig.3, Fig.4 and the text for Fig.3 on one page,

- What caused the measurement interruption on Fig.4a between Mar-20 to May-20? Then the statement on r.203 "We have performed 5-minute continuous measurements of PM10, PM2.5 ..." would need to be corrected (also for the measurement interruptions on Fig.3).

- "(around 0.3)" on r.313 do you mean PM2.5/PM10 ratio? - Please clarify this paragraph.

- r.364 states:: "... we performed the correlation analysis with n=556 days ..." . What is the correlation coefficient between PM10/PM2.5 and relative humidity (not a Coefficient of determination) in Fig.7/Fig.9?

- r.369 states, "The most common relative humidity value is 1% (22 times), followed by 74% (20 times)". The resulting 1% or 74% is the average, or Median within a day?

- For 1% or 74% and a selected other case of relative humidity, it would be interesting for a reader not living on the island to see what the pattern of change in relative humidity over 1 day is. Please document the changes in relative humidity in the figures.

- The authors note on r. 384 a "malfunctioning" of the sensor. How was the detected "malfunctioning" corrected ?

- What does "has been fairly stable" (r.431) mean, when in Fig.4a one can see a malfunction of almost 1,5 months and also other interruptions of measurements in Fig.3.

- Based on the information given in the article it is difficult to compare the measured values (but correlation is OK) at the measurement site with SDS011 and Vuelta de Los Pájaros, because: a reference instrument operating is 5 km away + 400 m in height + complex orography of the surrounding areas + one side is completely blocked by a wall. It would therefore be necessary to reconsider the suitability of e.g. Fig.6 and other information in the paper regarding the comparison of the magnitude of the measured values by the two sensors.

Author Response

We thank the referee for his/her detailed comments and suggestions. We address all of them below (bold text):

- In Fig.1, let the 3 lines be perpendicular, not inclined.

Thanks, we have improved Fig1a following your recommendations. We have also reordered the two panels, so that panel (a) is now on the left.

- What is the measurement accuracy of SDS011? Please also provide this information in ch.2.1, please.

We have added the accuracy provided by the manufacturer to line 133: “±15% of the measurement (minimum of ±10 µg/m3)”. In the same line, we have also fixed a typo in the units of the resolution.

- Does the design of the sensor require a PVC tube? Can the sensor be mounted directly in the air? I think the PVC tube may potentially distort the results in Fig.3.

Our sensor is integrated in the sensor.community network, which recommends using a 20 cm tube, see https://sensor.community/en/sensors/airrohr/. We have used a much shorter tube of 9 cm. Note the manufacturer only recommends the tube to be shorter than 1m, see e.g. https://cdn-reichelt.de/documents/datenblatt/X200/SDS011-DATASHEET.pdf The tube is also straight, so we believe particle losses by impact are not likely. 

In line 154 we have added “Note Sensor.Community recommends a 20 cm inlet [25], but we have used an even shorter one to minimize particle losses by impact.”

- How many power banks were needed for the measurement? 2 pcs or more?

Most of the experiment we alternated between two batteries, in line 168 we now mention that:

“Although at the beginning of the experiment we only used one battery, later we used two, so that while one battery was charging, the other was in use in the sensor.”

- What tool was used to automatically produce plots to check the daily evolution of the PM? 

All the scripts running in the Raspberry Pi 3 were coded by ourselves. To make it clear, line 171 now reads: “also run our own python scripts to gather the data from the monitor every 5 minutes and automatically produce plots”

- Describe in more detail the work with the Raspberry Pi 3 when measuring the PM.

As described in lines 173 to 176, the Raspberry Pi 3 had three functions: sending a warning when it could not connect to the sensor, downloading the data, and making plots. The SDS011 sensor itself produces the PM data.

- I recommend to put web links in References.

Thanks, we have followed your recommendation.

- What were the causes of the measurement failures with SDS011 on Fig.3?

We have added the following explanation on lines 260 to 268:

“Note there are some periods where data from the low-cost sensor are missing. This is specially the case at the beginning of the experiment, when we faced problems to keep the sensor continuously powered. As described in Section 2.1, this issue was solved using two batteries in rotation, and implementing a warning system in our Raspberry Pi. Other periods without data correspond to the malfunctioning of the humidity-temperature sensor. This was the cause for the ~1.5 months period without data from early March to mid April 2020. We latter decided to keep the SDS011 sensor running even when we were waiting for a replacement of the DHT22 sensor to arrive.”

- How long were the measurement interruptions with SDS011 (hours, days)?

The longest interruption corresponds to the ~1.5 months period mentioned above. Other interruptions were shorter, a few days at most.

- I recommend putting Fig.3, Fig.4 and the text for Fig.3 on one page,

The paper first presents the PM10 comparison with the reference sensor and this is shown in Fig 3. Then it discusses the SDS011 PM10 and PM2.5 measurements, and this is shown in Fig. 4. We believe having both Figs in one page in the first section would make the later section more difficult to follow.

- What caused the measurement interruption on Fig.4a between Mar-20 to May-20?

The DHT22 sensor broke and we waited for its replacement to arrive. We now mention this is in lines 264-268

Then the statement on r.203 "We have performed 5-minute continuous measurements of PM10, PM2.5 ..." would need to be corrected (also for the measurement interruptions on Fig.3).

We have changed the beginning of Section 2.3 to “We have performed 5-minute measurements of PM10, PM2.5, relative humidity and temperature with our low-cost sensors almost continuously during the period from November 10th, 2019 to May 28th, 2023 – 1020 days in total”

- "(around 0.3)" on r.313 do you mean PM2.5/PM10 ratio? - Please clarify this paragraph. 

Yes, it’s the PM2.5/PM10 ratio. Line 333 has been modified to “PM2.5/PM10 ratio of approx. 0.3”

- r.364 states:: "... we performed the correlation analysis with n=556 days ..." . What is the correlation coefficient between PM10/PM2.5 and relative humidity (not a Coefficient of determination) in Fig.7/Fig.9?

The correlation coefficients are very low: 0.20 and 0.15 for PM10 vs RH and PM2.5 vs RH, respectively. We have added these values to lines 397 (“Note that the Spearman correlation coefficient between PM10 and relative humidity is just 0.20.”) and 440 (“The Spearman coefficient between PM2.5 and relative humidity is again very low, 0.15.”)

- r.369 states, "The most common relative humidity value is 1% (22 times), followed by 74% (20 times)". The resulting 1% or 74% is the average, or Median within a day?

These are median values of daily data.

- For 1% or 74% and a selected other case of relative humidity, it would be interesting for a reader not living on the island to see what the pattern of change in relative humidity over 1 day is. Please document the changes in relative humidity in the figures.

We thank the referee for drawing our attention to this paragraph. We indeed believe our text was not clear, so we now provide the number of days (and its relative percentage over the total number of days) above relative humidity values of 70, 80 and 95, which are the thresholds found in the literature for the SDS011: line 391 now reads “Daily median values above 70% relative humidity happen  219 days, 39.4% of the total”, and line 395 includes “. In our dataset, values above the 80 and 95% relative humidity thresholds are found in 17.3% and 0.2% of the analyzed days, respectively”. For the PM2.5 dataset, the results are almost identical, and we mention it on line 437: “For the PM2.5 dataset, the number of days above the 70%, 80%, and 95% relative humidity is almost the same as for the PM10”.

- The authors note on r. 384 a "malfunctioning" of the sensor. How was the detected "malfunctioning" corrected ?

These are the same dates already found in Section 3.2.1 while discussing PM10, and as in that case, in PM2.5 analysis here we just remove those days from the dataset. In line 407 we now reference Section 3.2.1 to make this clear.

- What does "has been fairly stable" (r.431) mean, when in Fig.4a one can see a malfunction of almost 1,5 months and also other interruptions of measurements in Fig.3.

The sentence in the first version of our manuscript deals with the stability of the measurements when compared  to the reference instrument. In this sense, over all the 3.5 years of measurements, we only detect one noticeable shift, so we believe the instrument is fairly stable.

We now have added another sentence next to the previous one, see line 459: “Besides a 1.5 months period at the beginning of the experiment, our low-cost sensor has operated almost continuously over the 3.5 years period presented in this paper, so the instrument can be also considered fairly robust”

- Based on the information given in the article it is difficult to compare the measured values (but correlation is OK) at the measurement site with SDS011 and Vuelta de Los Pájaros, because: a reference instrument operating is 5 km away + 400 m in height + complex orography of the surrounding areas + one side is completely blocked by a wall. It would therefore be necessary to reconsider the suitability of e.g. Fig.6 and other information in the paper regarding the comparison of the magnitude of the measured values by the two sensors.

The comparison with the reference instrument is mainly performed to track changes in our low-cost sensor. In this sense, Fig 6 is used to detect the shift of the absolute error, and thus determine that some event affected our low-cost sensor.

Indeed, a direct comparison between the values of the reference and low-cost sensor should only be possible if both sensors were collocated and the SDS011 sensor had been calibrated with respect to the reference. As discussed in the conclusions, this is an experiment we have already planned.

To make this clear, by have added a new paragraph at the end of Section 2.2: “Note that the PM10 and PM2.5 values of the low-cost and reference sensors are not expected to be the same, because the two instruments are not collocated and their calibrations are presumably not related. Some degree of correlation between the data of both sensors can, however, be expected, because African dust outbreaks affect large areas. In this paper, we use the reference instrument to detect changes in the low-cost sensor, either over time or caused by the relative humidity. This is accomplished by looking at changes in the differences or in the correlations between both datasets. The reference instrument is also useful to help in the identification of the days affected by African dust.”

Reviewer 2 Report

The authors of the entitled manuscript “Long-term analysis of aerosol concentrations using a low-cost sensor: monitoring African dust outbreaks in a suburban environment in the Canary Islands.” investigated the operation of a low-cost sensor over a period three and a half years. I found the article is relevant to current air quality studies. I have the following points which I would like the authors to address:

1.       I found the introduction section did not align with the main aim of the journal, which is to illustrate the sensors’ function. Please find the following article which describes the function and set-up for a low-cost sensor.  M. A. Al-Rawi, P. Chand, and A. V. M. Evangelista, “Cost-Effective Customizable Indoor Environmental Quality Monitoring System”, Adv. technol. innov., vol. 7, no. 1, pp. 01–18, Oct. 2021.

2.       Aside from testing the sensor over a long period of time, what is the novelty of this work concerning the sensor?

3.       What is the difference between measuring PM2.5 and PM10 using aerosol sensors? Additionally, why is PM1 considered?

4.       How does the current sensor measure PM2.5 and distinguish it from PM10?

5.       I suggest that the authors add a Limitations section.

6.       What is the conclusion of this article? I found the current conclusion is just suggesting that we could use a low-cost sensor for long period, providing readings within an acceptable error.

Overall, this article has some merit, but in my opinion, it is not ready for publication given the current level of work and lack of novelty with respect to the sensor under investigation. 

Author Response

We thank the referee for his/her comments and suggestions. We address all of them below (bold text):

  1.   I found the introduction section did not align with the main aim of the journal, which is to illustrate the sensors’ function. Please find the following article which describes the function and set-up for a low-cost sensor.  M. A. Al-Rawi, P. Chand, and A. V. M. Evangelista, “Cost-Effective Customizable Indoor Environmental Quality Monitoring System”, Adv. technol. innov., vol. 7, no. 1, pp. 01–18, Oct. 2021.

We thank the referee for his/her comment. Although we give an overview of dust sensors in the Introduction, the discussion of the setup and function of the SDS011 in our paper is provided in Section 2.1, “Low-cost dust monitoring system”. Looking at the reference suggested by the referee, we believe we have included all the relevant information. However, it’s true that more information on the type of sensors could be mentioned to the Introduction, and thus we have added the following sentence in line 89 which includes a reference to the suggested paper: “Extra sensors can also be included to monitor other variables, such as temperature, relative humidity, UV radiation, and even ambient noise [19].”

  1.   Aside from testing the sensor over a long period of time, what is the novelty of this work concerning the sensor?

As mentioned in the Abstract and the Conclusions, our paper shows that the SDS011 can be used to track African dust outbreaks, which is a major problem in some areas of the World. It also shows that, in this case, it does not require special maintenance, making it suitable for citizen networks which could complement official networks. Of course, the fact that, for the detection of African dust, the sensor can be used for a much longer period than the one recommended by the manufacturer is also very important for this type of network.

  1.   What is the difference between measuring PM2.5 and PM10 using aerosol sensors? Additionally, why is PM1 considered? 

Particle size affects the scattering of light, thus making it possible for laser scattering sensors to distinguish the different PM fractions. PM10 and PM 2.5 are the most relevant particle sizes related to the detection of the coarse African dust. Very fine particles in the PM1 fraction are mainly associated with anthropogenic emissions. 

Line 80 now includes all this additional information: “Light scattering is affected by particle size, and thus these devices are capable of measuring different particle fractions including PM10, PM2.5, and PM1 – the former two are relevant for African dust studies, while the very fine PM1 is mainly associated with anthropogenic emissions”

  1.   How does the current sensor measure PM2.5 and distinguish it from PM10?

The SDS011 belongs to the family of laser scattering sensors, and thus can distinguish PM2.5 from PM10 thanks to the effect of particle size on the light scattering, as outlined above. Unfortunately, the manufacturer of the SDS011 does not provide specific information on the operating algorithm used by the sensor.

  1.   I suggest that the authors add a Limitations section.

We already summarize the limitations of our study in the Conclusions. The main limitation is that the reference instrument is located 5 km away from the low-cost sensor. Although this precludes an absolute comparison of the two datasets, it should be stressed that the reference sensor data is mostly used to track changes (over time or with relative humidity) in the low cost sensor. We have added a new paragraph at the end of Section 2.2 to make this clear: “Note that the PM10 and PM2.5 values of the low-cost and reference sensors are not expected to be the same, because the two instruments are not collocated and their calibrations are presumably not related. Some degree of correlation between the data of both sensors can, however, be expected, because African dust outbreaks affect large areas. In this paper, we use the reference instrument to detect changes in the low-cost sensor, either over time or caused by the relative humidity. This is accomplished by looking at changes in the differences or in the correlations between both datasets. The reference instrument is also useful to help in the identification of the days affected by African dust.”

It should also be noted that, as discussed in the conclusions, we plan to perform experiments with both instruments collocated.

  1.   What is the conclusion of this article? I found the current conclusion is just suggesting that we could use a low-cost sensor for long period, providing readings within an acceptable error. 

As presented in section 4, the main conclusion of our work is that the SDS011 sensor can be used to detect African dust outbreaks. Note that African dust outbreaks are not only of scientific interest, but also have a major impact on the lives of the citizens of the affected areas, as outlined in the Introduction.

Furthermore, we indeed also find that, for the particular task of measuring African dust, the sensor can operate for a long period (much longer than the lifespan provided by the manufacturer), with minimal maintenance and almost no interference of relative humidity. All this makes this sensor an ideal tool for large, semi autonomous networks to complement reference instruments.

Overall, this article has some merit, but in my opinion, it is not ready for publication given the current level of work and lack of novelty with respect to the sensor under investigation.

As discussed in the conclusions, we have indeed further work planned, including experiments with the SDS011 and reference sensors collocated. However, the present study is by no means short of effort -- it includes a wealth of data (the longest SDS011 time series to our knowledge, with all the work required to keep the sensor operative for 3.5 years) with a full analysis (which includes a comparison with a reference sensor using statistical methods, and an assessment of the dust outbreak events based on the official data provided by the Spanish METDC). This allows us to find novel results -- in particular, the feasibility of using the SDS011 sensor in African dust studies over long periods of time with little interference of relative humidity. Thus, we believe this is a complete study which can be useful to the scientific community. We hope all the additions and clarifications introduced during the paper revision following all the referees’ comments will improve the opinion of the present referee on our present study.

Reviewer 3 Report

The research deals with data collection using standard commercial sensors and a statistical analyzer for monitoring African dust outbreaks in a suburban environment in the Canary Islands.
Standard methods are used for data collection and processing. It is interesting to compare a Low-cost dust monitoring system with the reference monitor.

-It would be better to redraw Figure 1 (a).

-Has the effect of sensor lifetime on the accuracy of measured data been investigated?
-What methods could be used to reduce electricity consumption?
-Has the effect of sensor placement been studied?
-The studied and reference sensors should have the same location.
-Has the effect of sensor placement been investigated?
-Why was Spearman coefficient chosen?

Author Response

We thank the referee for his/her useful comments and suggestions. We address all of them below (bold text):

-It would be better to redraw Figure 1 (a).

We have updated Fig 1 (a). We have also reordered the two panels of Fig1, so that panel (a) is now on the left.

-Has the effect of sensor lifetime on the accuracy of measured data been investigated?

The comparison with the reference sensor, even though it’s not operating at the same  place as the low-cost one, has allowed us to investigate the stability of the sensor measurements over time. This way we have found the shift in the MAE discussed in Sections 3.2.1 and 3.2.2 for PM10 and PM2.5 data, respectively. The manufacturer provides an accuracy of ±15% of the measurement, with a minimum of ±10 µg/m3 – we have now included this data in Section 2.1. Measurements with a collocated reference instrument with a known accuracy could be used to check this value and to study its evolution over time.

-What methods could be used to reduce electricity consumption?

We considered two methods during our experiment: switching from wifi to some other low power wireless transmission method (such as LoRa or bluetooth low energy), and switching the Node MCU to a low power state when not measuring (currently, the Node MCU is always on, including the wifi connection). We have included the following lines at the end of the conclusions: “In order to improve the autonomy of the sensor and reduce even more the need for human intervention, we would also like to investigate methods to reduce energy consumption,  such as switching the Node MCU to a low power state between measurements, or using LoRa or Bluetooth Low Energy technologies for data transmission.”

-Has the effect of sensor placement been studied?

In our current study, we replicate the conditions of a citizen-scientist experiment, so the sensor is placed in a window. Although there is an overall good agreement with the reference instrument, this placement likely has some effect on the measurements. To ascertain this effect we could perform an experiment running two low-cost sensors at the same time in close positions but with slightly different placements. We now include this specific experiment among the ones already discussed in the conclusions, see line 483: “This type of experiment would also allow us to check the effect of sensor placement on the measurements.”

-The studied and reference sensors should have the same location.

Indeed, the reference instrument is located 5 km away from the low-cost sensor. Although this precludes an absolute comparison of the two datasets, it should be stressed that the reference sensor data is mostly used to track changes (in time or with relative humidity) in the low cost sensor. We have added a new paragraph at the end of Section 2.2 to make this clear: “Note that the PM10 and PM2.5 values of the low-cost and reference sensors are not expected to be the same, because the two instruments are not collocated and their calibrations are presumably not related. Some degree of correlation between the data of both sensors can, however, be expected, because African dust outbreaks affect large areas. In this paper, we use the reference instrument to detect changes in the low-cost sensor, either over time or caused by the relative humidity. This is accomplished by looking at changes in the differences or in the correlations between both datasets. The reference instrument is also useful to help in the identification of the days affected by African dust.”

Note that, discussed in the conclusions, we have already planned an experiment with the low-cost sensor collocated with a reference instrument. 

-Has the effect of sensor placement been investigated?

See the two questions above.

-Why was Spearman coefficient chosen?

As mentioned in line 239, we favor the Spearman over the Pearson correlation coefficient to reduce the effect of outliers on the results.

Round 2

Reviewer 1 Report

The authors accepted most of the comments, but some comments were not accepted by the authors, moreover, I am adding more of my questions and comments:

 - In Fig.1, the 3 lines (black, green and blue) leading to SDS 011 should be straight, not tilted.

- How long were the measurement interruptions with SDS011 in Fig.3 and what caused them? (besides the 1.5 month interruption already explained)

- On r.407 authors stated: "likely due to the malfunctioning of either one of the dust monitors." How many sensors were used for the measurement?

- On r.313, 315, 334, 350 and 408 the authors mention malfunctioning of the sensor. How did you correct the malfunction? I get the impression from the article that on one day the sensor had a malfunction, you did not intervene, and continuously on the next day the sensor was already measuring correctly...

- I recommend to put Fig.3 and the text related to Fig.3 on one page. Similarly with Fig.4.

- On p.398 there is a mention of Spearmon's correlation coefficient. Please, calculate Pearson's correlation coefficient as well.

Author Response

We thank the referee for his/her comments, which we address below (see bold text) and in the manuscript (see text highlighted in green):

 - In Fig.1, the 3 lines (black, green and blue) leading to SDS 011 should be straight, not tilted.

We are truly sorry – we actually fixed the image the last time, but did not update it in the revised manuscript. Hopefully it will be OK now.

- How long were the measurement interruptions with SDS011 in Fig.3 and what caused them? (besides the 1.5 month interruption already explained)

Beside the DHT error of 1.5 months, the measurement interruptions were caused by lack of electrical power – before we implemented the warning system, the sensor went without power for periods of up to ~9 days, which were shortened to 1-2 days in most cases once the warning system was operational. The beginning of Sec 3.1.1 now reads: “This is specially the case at the beginning of the experiment, when we faced problems to keep the sensor continuously powered, causing data gaps of up to approx. 9 days, although usually of just 1-2 days. As described in Section 2.1, this issue was solved using two batteries in rotation, and implementing a warning system in our Raspberry Pi, which helped us to shorten the data gaps to 1-2 days in most cases.”

- On r.407 authors stated: "likely due to the malfunctioning of either one of the dust monitors." How many sensors were used for the measurement?

In Section 3.2.2 we are comparing the data of two sensors: the low-cost one and the reference monitor at Vuelta de los Pajaros. To make it clear, we have changed the sentence to “likely due to the malfunctioning of either the low-cost sensor or the reference dust monitor”

- On r.313, 315, 334, 350 and 408 the authors mention malfunctioning of the sensor. How did you correct the malfunction? I get the impression from the article that on one day the sensor had a malfunction, you did not intervene, and continuously on the next day the sensor was already measuring correctly…

We didn’t do anything to the sensor when these malfunctions took place – we only detected them later when doing the comparison with the VP reference monitor. Since the sensor continued measuring correctly afterwards, we believe that these were some exceptional (note this only happens 4 days) errors, for example while reading the data or transmitting it. At the end of Section 3.1.1, while first discussing these errors, we now mention possible causes: “these peaks could be due to some malfunctioning of our low-cost sensor (for example, a data read, or transmission, error)”.

Note that, as already mentioned in the text, we also can’t exclude the possibility of some other source (likely of anthropogenic origin) causing these high PM values.

- I recommend to put Fig.3 and the text related to Fig.3 on one page. Similarly with Fig.4.

Yes, we would also like to see all three panels of Fig 3, plus its figure caption, in one single page. However, we don’t believe we should do this at this point – we think this will be done by the journal’s technical staff at a later stage, but we have consulted the editor. Note we have followed the authors’ guidelines and placed the figures as close as possible to where they are first mentioned in the text.

- On p.398 there is a mention of Spearmon's correlation coefficient. Please, calculate Pearson's correlation coefficient as well.

We have added the Pearson coefficients for the PM10 and PM2.5 vs RH correlation calculations to the text: “Note that the Spearman correlation coefficient between PM10 and relative humidity is just 0.20 (Pearson coefficient of 0.093)” and “The Spearman coefficient between PM2.5 and relative humidity is again very low, 0.15 (Pearson coefficient of 0.094)”

In the rest of the paper, we always provide both the Spearman coefficient and the r2 coefficient of determination, the latter just being the square of the Pearson coefficient for simple linear regressions such as the ones presented in this paper.

Reviewer 2 Report

I would like to thank the authors for answering all my questions. In my opinion, the current revised version is ready. 

As an indoor air quality researcher and to the best of my knowledge, no one has done a long time study in the field. For future work, I will suggest to the authors to use the data with machine learning theories as it will be a good article. 

Author Response

We thank the referee for his/her positive comments on our research. We are indeed very interested in the application of machine learning methods for data classification and forecast.

Round 3

Reviewer 1 Report

I would therefore like to inform you that the authors, for the second time, have not responded to my suggestions regarding the 24 hour humidity time course. I consider this an important part of the article to be added. They have corrected everything else, but have not addressed this issue.

Author Response

We thank the referee for this remaining comment and all the previous ones. However, there must be some kind of confusion, because the only comments that reached us regarding relative humidity were made in the first round, and we addressed them all. In the second round, there were no comments regarding relative humidity, so we assumed we had successfully addressed the referee’s questions on this topic.

Regarding the 24 hour values, the referee’s question and our answer in round 1 were:

>>>

- For 1% or 74% and a selected other case of relative humidity, it would be interesting for a reader not living on the island to see what the pattern of change in relative humidity over 1 day is. Please document the changes in relative humidity in the figures.

We thank the referee for drawing our attention to this paragraph. We indeed believe our text was not clear, so we now provide the number of days (and its relative percentage over the total number of days) above relative humidity values of 70, 80 and 95, which are the thresholds found in the literature for the SDS011: line 391 now reads “Daily median values above 70% relative humidity happen 219 days, 39.4% of the total”, and line 395 includes “. In our dataset, values above the 80 and 95% relative humidity thresholds are found in 17.3% and 0.2% of the analyzed days, respectively”. For the PM2.5 dataset, the results are almost identical, and we mention it on line 437: “For the PM2.5 dataset, the number of days above the 70%, 80%, and 95% relative humidity is almost the same as for the PM10”.

>>>

As explained in our answer to round 1 above, we believe the values provided for the number of days above the usual humidity thresholds are sufficient for the objective of this paper, because we’re analyzing PM daily data and using these relative humidity values in Sections 3.2.1 (see fig 7 in particular) and 3.2.2 (fig 9)

Indeed, we have the relative humidity data for almost 600 days and with a 5 minute resolution, but we don’t believe showing some statistical evolution over a 24 hour period would provide additional information to this study, which deals with daily medians. Note in particular that all PM figures show daily data over almost 3.5 years, so it’s not possible to plot the humidity in them with a higher resolution.

However, we thank the referee for drawing our attention to this topic and, if in the future we analyze a shorter period and use higher resolution (e.g. hourly instead of daily), we will of course provide suitable relative humidity plots with a high time resolution.